# The Plant Volatile-Sensing Mechanism of Insects and Its Utilization

**DOI:** 10.3390/plants13020185

**Published:** 2024-01-10

**Authors:** Qi Qian, Jiarong Cui, Yuanyuan Miao, Xiaofang Xu, Huiying Gao, Hongxing Xu, Zhongxian Lu, Pingyang Zhu

**Affiliations:** 1College of Life Sciences, Zhejiang Normal University, Jinhua 321004, China; qianq5119@163.com (Q.Q.); 2019202036@njau.edu.cn (J.C.); m13506637616@163.com (Y.M.); ghy2029@163.com (H.G.); luzxmh@163.com (Z.L.); 2State Key Laboratory for Managing Biotic and Chemical Threats to the Quality and Safety of Agro-Products, Institute of Plant Protection and Microbiology, Zhejiang Academy of Agriculture Sciences, Hangzhou 310021, China; 3Jinhua Agricultural Technology Extension and Seed Administration Center, Jinhua 321017, China; 13665895587@163.com

**Keywords:** plant volatiles, insect, olfactory system, attractant, pest management

## Abstract

Plants and insects are engaged in a tight relationship, with phytophagous insects often utilizing volatile organic substances released by host plants to find food and egg-laying sites. Using plant volatiles as attractants for integrated pest management is vital due to its high efficacy and low environmental toxicity. Using naturally occurring plant volatiles combined with insect olfactory mechanisms to select volatile molecules for screening has proved an effective method for developing plant volatile-based attractant technologies. However, the widespread adoption of this technique is still limited by the lack of a complete understanding of molecular insect olfactory pathways. This paper first describes the nature of plant volatiles and the mechanisms of plant volatile perception by insects. Then, the attraction mechanism of plant volatiles to insects is introduced with the example of *Cnaphalocrocis medinalis*. Next, the progress of the development and utilization of plant volatiles to manage pests is presented. Finally, the functions played by the olfactory system of insects in recognizing plant volatiles and the application prospects of utilizing volatiles for green pest control are discussed. Understanding the sensing mechanism of insects to plant volatiles and its utilization will be critical for pest management in agriculture.

## 1. Introduction

Insects and plants have co-evolved for hundreds of millions of years [1]. During this evolutionary process, insects have developed a unique olfactory system that distinguishes plant volatiles from other environmental odors [2,3]. Plant volatile organic chemicals (PVOCs) are essential chemical information links for insect-searching host plants and locating habitat, and play critical roles in the interdependent relationship between plants and insects. Insects can use these volatile compounds to collect information about plants, such as assisting bark beetles in the location of stressed host trees [4]. Research on the olfactory sensory mechanism of insects on PVOCs can help reveal the coevolutionary relationship between insects and plants and provide a theoretical basis for developing ecological technologies for preventing and curing pests.

Plant volatiles meet several key prerequisites for modern pest management, including being species-specific and environmentally benign [5]. Methods that interfere with an insect’s normal sense of smell fulfill these conditions and have been implemented on a large scale in the field. For example, physicochemical trap technology utilizing insect pheromones and plant volatiles to attract pests is highly targeted and provides efficacious pest management, reducing the need for traditional chemical applications [6,7]. As modern molecular and behavioral biology techniques are applied to further study the interactions of plant volatiles on pest behavior, the capabilities of olfaction-based pest management will continue to advance. High-throughput screening methods are particularly promising and will enable the identification and testing of highly efficacious, natural plant volatiles to alter the behavior of or trap agronomic pests. This has the potential to continue to improve our pest management technology in a sustainable and environmentally friendly manner in line with modern goals of ecological agriculture [8,9,10,11].

In this review, we employed NCBI PubMed to summarize the progress in comprehending how insects recognize and utilize plant volatiles over the past decade. We first describe the plant volatile species and their properties and the response of phytophagous insects to plant volatiles. Then, we list the relevant protein species of the olfactory system in the antennae of a primary lepidopteran pest of rice, *Cnaphalocrocis medinalis*, and their functions. Finally, we look forward to the latest research ideas and methods in insect chemical ecology research to provide a theoretical basis for the green control of pests.

## 2. Plant Volatiles

PVOCs are a mixture of multiple volatile plant secondary metabolites. Plant volatiles can be divided into green leaf, floral, and fruit volatiles according to the different organs of the plant, in which straight-chain alcohols and aldehydes containing six carbon atoms as well as their esters are the primary source of green leaf odors. Terpenes such as monoterpenes, sesquiterpenes, and sesquiterpenes together with aromatic compounds are the main constituents of the floral odors of the plant. The short-chained acetic esters formed by the degradation pathway of carbohydrates are the main source of the fruit aroma (Table 1). The metabolic pathways of different plants can produce specific odors, such as the cystine and cysteine metabolic pathways of the lily family and the methionine metabolic pathway of the cruciferous family [12,13].

The production of PVOCs by plants, including both the diversity and abundance of the compounds, is strongly influenced by biotic and abiotic factors [14,15], and they also play essential functions in plant–insect interactions. The study of PVOCs is complicated by the number of compounds involved and the complexity of the interactions that they facilitate. For example, multiple PVOCs with different functions often originate from the same plant tissue or organ, all of which can be involved in interactions with numerous insect species. In just flowers, Martel et al. [16] found that not only do floral cuticle hydrocarbons play an important role in plant–pollinator interactions, but these same hydrocarbons are also critical for interspecies communication among insects.

PVOCs directly affect insect mating and reproductive behavior and help insects search, locate, and select suitable host plants. Different classes of PVOCs have different attractant properties to insects. Pests often use plant green leaf odor to locate hosts, while floral and fermented molasses odors are the main signals for non-pest insects to locate pollen, nectar, and other food sources. However, these are generalizations and not rules. For example, the floral scent substances attractive to Lepidoptera insects include phenylacetaldehyde, β-geranylene, and methyl salicylate. However, phenylacetaldehyde is also used by a variety of Lepidoptera pest species to locate suitable hosts [17]. Additionally, Feng et al. [18] found that floral odor, color, and nectar secretion would change independently throughout flowering in *Lonicera japonica*, suggesting that the combination of both visual and olfactory cues may play a role in attracting or filtering different visitors.

Arthropod pests evolve resistance to insecticides at varying rates, averaging only 60–78 generations [19]. Therefore, there is a need to find safer and more reliable methods to achieve sustainable pest control. Using volatile signals between plants and pests can be part of this more holistic and ecological approach to pest management. Yan et al. [20] found in field experiments that *trans*-2-hexen-1-ol and isopropyl isothiocyanate from cruciferous plants were highly attractive to *Plutella xylostella* adults, especially when used in conjunction with yellow and green sticky traps. Field trials confirmed the synergistic effect of phenylacetaldehyde and linalool on the trapping of *Anticarsia gemmatalis* [21]. In conclusion, although many monomers or simple formulations of plant volatile attractants have been reported [22,23], the ratio of pest population control, as well as the selectivity of attraction to target pests and beneficial insects, still needs to be improved. The following problems are highlighted: First, although several studies on field screening of pest-regulating PVOCs have been carried out, the scope of field trials is necessarily limited to the systematic and comprehensive screening of plant volatiles. Second, background levels of green leaf odorants can interfere with deployed green leaf odorants, thus limiting the effectiveness of trapping and plant masking. Third, the unknown mechanisms of pest attraction and the variable effects of PVOCs complicate screening. Clarifying the molecular mechanisms underlying the attraction of pests by plant volatiles will help improve the speed and reliability of PVOC screening.

## 3. Molecular Perception of PVOCs by Insects

### 3.1. The Role of Insect Antennal Olfactory Sensors in the Recognition of PVOCs

Insect antennae sensors have more than ten types based on the structural characteristics of the sensilla’s epidermis and its mode of attachment [24]. Different kinds of sensilla have their own roles, e.g., the sensilla squamiformia is a receptor that senses mechanical stimuli [25]; the sensilla trichodea is associated with plant volatile recognition [26]; the sensilla chaetica is associated with taste [27]; and most sensilla basiconca are a class of olfactory receptors which can function in capturing plant volatile molecules using a large number of pore structures on their surfaces which contain a large number of neuronal cells (Figure 1) [28]. 

Insects use special olfactory sensors to capture hydrophobic lipid-soluble small molecules in the air. Olfactory receptors can then differentiate between many complex odorant molecules and generate olfactory signals [29], which regulate behaviors such as feeding, courtship, and defense against predators [10,30]. Insect olfactory sensors, the main organs insects use to perceive the outside world, are overwhelmingly distributed in their antennae [31]. These olfactory sensors are in the form of sensilla, which exist in different forms according to their function. For example, the sensilla trichodea in the antennae of *C. medinalis* may play an important role in host localization, whereas the sensilla basiconca exists in a sexually dimorphic pattern and plays different roles in olfactory perception in male and female adults [32]. 

Insect olfactory receptors show similarities between phylogenetically related species. For example, the type and distribution of larval antennal receptors in two Zygoptera species, *Ischnura elegans* and *Calopteryx haemorrhoidalis*, are similar to those of other species in the order [33]. However, the number, type, and localization of sensilla vary significantly among species. *Callosamia promethea* have more than 60,000 sensilla trichodea on their antennae, while *Choristoneura fumiferana* have only 2300 [34]. Sensilla campaniform are present only in the antennae of male *Hepialus yulongensis* and not the females [35].

Currently, the dominant research method for constructing insect olfactory sensors is electron microscopy [36,37]. With the updating of technical means, the ultramicrostructure of the sensory organs on the antennae of insects continues to become clearer [38,39,40]. Using scanning and transmission electron microscopy techniques, Sun et al. [32] observed eight morphological types of the sensilla of *C. medinalis*: sensilla trichodea, sensilla basiconc, sensilla coeloconica, sensilla styloconica, sensilla squamous, sensilla auricillica, Böhm bristles, and sensilla cavity. Du et al. [41] combined scanning electron microscopy techniques with electroantennography to discover alarm pheromone-specific receptors on the antennae of *Aphis glycines*. However, structural analysis of olfactory sensors is important; ultimately, it is molecular interactions that govern odorant detection.

### 3.2. The Role of Insect Antennal Olfaction-Related Proteins in the Recognition of PVOCs

Odor molecules enter the receptor lymphatic fluid through micropores in the olfactory sensory epidermis. However, hydrophobic odorants cannot cross the hydrophilic lymphatic fluid to reach the odorant receptor neuron (ORN) and must be carried by a transport protein. These odorant-transporting proteins fall into two categories known as odorant-binding proteins (OBPs) or chemosensory proteins (CSPs) [10]. After being wrapped by a transport protein in the lymphatic fluid to form a complex, the odorant is transported to the ORN and activates the odorant receptors (ORs), ionotropic receptors (IRs), or sensory neuron membrane proteins (SNMPs) on the dendritic membrane. Subsequent excitation of the olfactory neurons is induced, converting chemical signals into electrical signals, which are transmitted as action potentials to higher nerve centers (i.e., antennal lobes and mushroom bodies). Finally, the higher nerve centers integrate the electrical signals and release nerve impulses that direct the insect to produce specific physiological and behavioral responses (Figure 2) [10,42,43].

OBPs are a class of small molecule water-soluble proteins with molecular weights of about 15–17 kDa and a signal peptide of approximately 20 amino acids at the N-terminus, which are present in high concentrations in the lymphatic fluid of receptors [44]. Insect OBPs generally contain six conserved cysteines that form three disulfide bonds for supporting and maintaining the stability of the protein structure and have hydrophobic binding cavities formed by folding six α-helical structures. Due to their small molecular weight, high water solubility, good stability, ease of modification, and ease of obtaining in vitro, OBPs were one of the first classes of carrier proteins to be studied in insects. In addition, they are also an essential class of lymphatic olfactory proteins, and their functions are best known due to their ease of study [45,46]. Comprehensive studies have shown that insect OBPs perform several physiological functions in both olfactory and non-olfactory tissues, including (1) transporting plant volatiles to receptor proteins, (2) assisting in activating the receptor, (3) transporting and releasing insect pheromones, (4) protecting PVOCs from degradation by odorant-degrading enzymes during transport, (5) degrading high concentrations of PVOCs to avoid excessive stimulation of olfactory neurons, (6) removing extraneous substances from the sensory lymph fluid, (7) participating in insect physiological development and tissue regeneration, (8) acting as blood anticoagulants in blood-sucking insects, (9) playing a role in the development of drug resistance in insects, and (10) participating in insect feeding and nutrient uptake [44,47,48].

The genes for *OBPs* exercising olfactory functions are highly expressed in the antennae [49]. With the wide application of various bioinformatic techniques, the research on OBPs has developed rapidly. The number of odorant-binding proteins identified in insects continues to increase [10,50,51]. For example, more than 507 genes encoding OBP in Lepidoptera, 102 in Hymenoptera, 8 in Homoptera, 104 in Hemiptera, 299 in Diptera, 295 in Coleoptera, and 48 in Blattodea have yet to be identified [52]. At present, some progress has been made in understanding the binding mechanism of OBPs through crystal structure analysis [53,54,55]. It has been found that the C-terminus fragments of some OBPs can control ligand binding and release using pH-dependent conformational changes [56,57]. Additionally, further investigation of the binding mechanisms between OBPs and their ligands was conducted using homology modeling combined with molecular docking technology to create 3D protein models [58,59]. Based on this technique, several key binding sites in the binding cavity of OBPs were found. For example, in the OBP McinOBP4 of *Macrocentrus cingulum,* the Met119 site forms a hydrophobic bond with the functional group of limonene, *trans*-3-hexen-1-ol-acetate, which can then be transported in the lymph [60]. In the OBP HarmPBP1 of *Helicoverpa armigera*, the sites Phe12, Trp37, and Phe119 may be involved in the binding to the main components of the insect’s sex pheromones, *cis*-11-hexadecenal and *cis*-9-hexadecenal [61]. 

Another important class of olfactory protein is the chemosensory proteins (CSPs), which are small, compact, soluble polypeptides. CSPs typically consist of 100 to 120 amino acids with molecular weights of about 10 to 15 kDa. These proteins bind and transport hydrophobic information chemicals. CSPs across different species of insects contain four conserved cysteines, forming two disulfide bonds that maintain the stability of the three-dimensional protein structure. This represents a decrease in conserved cysteines compared to OBPs [10]. 

Olfactory receptors (ORs), critical to insect olfaction, are generally composed of 300–350 amino acid residues and are a class of membrane proteins located on ORNs. ORs can be divided into two main groups: (1) the common odorant-binding protein (ORx), which is highly differentiated across species of insects, and (2) the odorant co-receptor (ORco), which is highly conserved across different insect species. Although ORcos do not directly bind odor molecules, ORcos assist ORxs in binding specificity and recognizing the correct odorants [62]. ORcos represent one of several classes of olfactory proteins critical to the functioning of insect olfactory systems but do not bind directly to odorant molecules, such as SNMPs and IRs. SNMPs are insect-specific membrane proteins comprising approximately 520 amino acids and are highly expressed in antennae. SNMPs are thought to contribute to insect odor recognition and are divided into three subfamilies (SNMP 1–3) [63]. IRs contain two or more co-receptors, where each co-receptor is co-expressed with one or two other IRs. These IRs have complex interactions and work together to sense acids, amino acids, and other compounds. However, studies on the functions and mechanisms of IRs have primarily been conducted on *Drosophila*, and data from other insects, such as Lepidoptera, are rare [64]. Insect olfactory sensitivity is closely related to the expression of several types of olfactory proteins. These proteins determine the recognition of different odors through specific binding sites and co-factors that regulate the sensitivity of the olfactory system to specific odorant molecules. Through new bioinformatics technologies, the research on olfactory proteins has been developing rapidly [65]. This in-depth knowledge of the mechanisms of the insect olfactory system plays a crucial role in exploiting insect attraction with plant volatiles for agronomic purposes.

## 4. The Attraction Mechanism of Plant Volatiles to Insects: A Case Study of *C. medinalis*


As the primary lepidopteran pest of rice, *C. medinalis* has become a model organism for studying the mechanisms of OBPs and CSPs to detect odorants. With the rapid development of sequencing technology, the database of *C. medinalis* is constantly growing: Zeng et al. [66] identified genes coding for 30 *OBPs*, 35 *CSPs*, 29 *ORs*, 15 *IRs*, and 2 *SNMPs* from the transcriptomes of *C. medinalis*; Liu et al. [67] identified 22 candidate *CSP* genes from the transcriptome of *C. medinalis*; and Liu et al. [63] identified genes coding for 12 *OBPs*, 15 *CSPs*, 46 *ORs*, 15 *IRs*, and 2 *SNMPs* from the antennae of *C. medinalis* using transcriptome sequencing.

To investigate the distribution of odor-binding proteins and the relationship with adult age and sex, antennal, genital, and leg tissues of *C. medinalis* adults of different sexes and ages were analyzed. Interestingly, 12 *OBP* genes were exclusively expressed in antennae. In the antennae, there were significant differences between sexes and among instars for expressing *OBPs* and *CSPs*. Moreover, the mating status of the insects did not affect the expression level of *OBPs* in the antennae of female moths [68]. Among the *CSP* genes, after analyzing the expression profile of the *CSP* genes with qPCR, it was found that *CmedCSP4*, *CmedCSP8*, *CmedCSP11*, *CmedCSP18*, and *CmedCSP21* were mainly expressed in the antennae [67].

Understanding the binding ability of odorant proteins to compounds in the environment is important for the development of targeted traps. In recent years, the binding ability of analyzed proteins to different ligands has been studied mainly with fluorescence-binding assays. Sun et al. [69] found that *CmedPBP4* in *C. medinalis* was specifically produced in the trichome sensilla of adult antennae, and the expression level of male adults was higher than that of females. Under different pH conditions, CmedPBP4 had a differentially high binding ability with four insect sex pheromone components, such as (*Z*)-11-hexadecenal, and with 11 rice PVOCs, such as cyclohexanol, nerolidol, and cedrol. These results indicate that CmedPBP4 may be a promising target olfactory protein for interfering with the mating of *C. medinalis* and, thus, its management. However, these experiments have yet to be conducted. Sun et al. [70] found that CmedOBP14 had a high binding affinity to 26 rice PVOCs such as cedrol, β-ionone, nerolidol, 3-carene, and 1-octen-3-ol. In addition, the authors speculated that *L*-limonene-mediated attractive activity may involve multiple OBPs or CSPs. Duan et al. [71] found that CmedCSP33 had a high binding affinity to seven kinds of rice PVOCs, such as 2-heptanone, β-ionone, cyclohexanol, (*E*)-2-hexen-1-ol, 3-pentanol, nerolidol, and *R*-(+)-limonene. CmedCSP33 also demonstrated a change in competitive binding affinities under different pH conditions. Further studies based on ligand binding assays, fluorescence quenching analysis, and circular dichroism (CD) showed that nerolidol and β-ionone influence insect behavior. The chemosensory proteins CmedCSP1 and CmedCSP2 of *C. medinalis* were found to have broader binding affinities than other closely related proteins and could bind with the most common volatiles of rice [72]. CmedCSP1 had the strongest binding affinity to terpenes, while CmedCSP2 bound most strongly to both long-chain alkanes and terpenes. In contrast, CmedCSP3 had more specific binding characteristics, binding most strongly with cyclohexanol and terpenes in plant green leaf volatiles. Though exploration is constantly updated, the molecular binding ability of some odor-binding proteins with specific odors has been clarified one after another, which provides an essential reference basis for screening out specialized odor substances.

Currently, 15 *OBP* genes, 15 *CSP* genes, 46 *OR* genes, 20 *IR* genes, and 2 *SNMP* genes have been identified from the antennae of *C. medinalis* [63,70]. While identifying olfactory-related proteins has laid the foundation for subsequent protein function analysis, presently, only a few proteins have undergone such a functional analysis. CmedPBP4, CmedOBP14, and CmedCSP33 have been analyzed for their molecular mechanism in PVOC recognition. Though, as of yet, the complete pathway of PVOC recognition in *C. medinalis* has yet to be fully elucidated. This lack of functional analysis of identified proteins represents a continued barrier to PVOC implementation in agronomic settings and presents an opportunity for further study (Table 2).

## 5. Development and Utilization of Pest Attractants Based on PVOCs

Through long-term coevolution, insects and plants have formed a close relationship. For example, volatiles released by plant floral organs can attract a variety of lepidopteran adults, which plants require for gene exchange, while most lepidopteran adults also need to feed on plant nectar to obtain required nutrients [73]. Leveraging this close relationship provides an effective way to develop attractants by studying the effects of PVOCs on insect behavior and then screening components with high behavioral activity levels against pests.

Insect recognition of pheromones varies with background odor, and host plant volatiles enhance the stimulatory effects of sex pheromones in insects [74]. Grapevine PVOCs can strengthen the recognition of sex pheromones among *Lobesia botrana* [75]. Also, when the host PVOC, linalool, acted simultaneously with the main components of the *Helicoverpa zea* sex pheromones, *cis*-3-hexenol and *cis*-11-hexadecenal, the insect’s ORN released more intense pulses than when exposed to *cis*-11-hexadecenal alone [76]. In contrast, background odors can also reduce insect response to pheromones. For instance, the linden flower volatile heptanol is very attractive to adult *Agrotis ipsilon*. While heptanol is present in the environment, higher concentrations of sex pheromone stimulation are required to elicit a response in *A. ipsilon* [77]. Thus, key PVOCs can be utilized and combined with a matrix that can stabilize the release of a variety of volatile compounds that can be used to disrupt mating or to attract and kill pests. The current overuse of chemical pesticides can seriously jeopardize human health and the environment, and plant-derived chemoavoidants and insect attractants may provide solutions for this enormous agricultural problem.

In recent years, insect attractants developed based on plant volatiles have gradually being used for field pest control (Figure 3). Using funnel traps and floral volatiles, including methyl salicylate, phenylacetaldehyde, and eugenol, diluted in *n*-hexane as attractants and replaced every two weeks can trap *Cydalima perspectalis* adults [78]. Meagher [79] used different floral volatile treatments, including phenylacetaldehyde, benzyl acetate combined with phenylacetaldehyde, benzyl acetate, and benzaldehyde, as bait with standard Universal Moth Traps, ‘Unitraps’, with insecticide strips to kill moths captured in peanut fields. These odorants successfully trapped *Pseudoplusia includens*, with female moths accounting for more than 67% of those captured. In addition, the odor of *Buddleja davidii* can attract a variety of noctuid moths. Studies have found that *B. davidii* volatiles, including benzaldehyde, 4-oxoisophorone, α-farnesene, dihydrooxyisophorone, and isophorone oxide, may be effective in the development of moth attractants [80].

While a single compound can provide insect-trapping effects, synthetic attractant mixtures with PVOCs and sex pheromones can enhance trapping effects. For example, an odorant mix caused strong electrophysiological responses in the antennae of unmated male *Plutella xylostella* [81]. In a Y-tube olfactometer test, the attractiveness to male adults of the mixed treatment of host plant volatiles and sex pheromones was significantly higher than that of sex pheromones alone, leading to a higher capture rate. In another example, the plant volatile *trans*-2-hexenyl acetate enhanced the trapping effects of the synthetic *Holotrichia paralleal* sex pheromone [82]. PVOCs can also interfere with insect sex pheromone detection. Molecular investigations revealed that when *Heliothis virescens* received sex pheromones in the presence of PVOCs, a calcium signal interfered with normal information transmission [83]. In addition, changes in the configuration of odorant molecules can also change the effectiveness of a PCOV on insects. For example, linalool exists in nature in the form of two enantiomers which have entirely different functions. (*S*)-(+)-linalool mainly attracts pollinators, and (*S*)-(−)-linalool seems to have the effect of repelling insects [84].

## 6. Conclusions

PVOC-mediated insect–plant interactions are the subject of intense research. Specifically, the behavioral responses of insects, synthesis, modes of action, mechanisms of insect olfactory system detection, and application in integrated pest management are all areas of current PVOC research interest [85,86,87]. Utilizing analytical chemistry, chemical ecology, neurophysiology, bioinformatics, and molecular biology techniques to better understand insect mechanisms of PVOC perception will continue to improve the effective use of PVOCs in agricultural pest management. Research into the molecular mechanisms of insect olfactory systems may allow for the computational analysis of the mechanisms of chemical communication between insects and plants. In molecular biology, extensive research is being conducted on insect olfactory receptors. This research has been especially focused on PVOC-binding proteins, sensory-binding proteins, and chemical signal transduction mechanisms involved in insect PVOC detection signaling [62,88]. In-depth identification of insect genes involved in the detection, and behavioral response to, PVOCs will provide the theoretical basis for the precise development of pest attractants [89]. This may offer a new way to screen PVOCs for attractant or repellent activity with much higher throughput than traditional screenings using ecological, electrophysiological, and behavioral methods. Increased screening throughput will provide new and more effective products for use in the sustainable management of pests [9,23,46].

While homology modeling and molecular docking can be utilized to predict the interaction between insect odorant-binding proteins and PVOCs, further exploration and verification through experiments are still required. In addition, it is not clear whether chemosensory proteins are also involved in the binding of PVOCs. Further research is needed to determine which ORs, IRs, or SNMPs on the dendritic membrane are activated and cause sensory responses following odorant molecule transport to the ORN. Studying pest-attractive PVOCs at the molecular level will help to clarify the molecular mechanism behind attractant recognition and determine which binding functions are critical to the process. A deeper understanding of the molecular process of attractant binding will help to optimize the type, concentration, and ratio of odorants needed to efficaciously trap pests such as *Cnaphalocrocis medinalis*. 

Compared to the vast array of volatile compounds plants produce, relatively few PVOCs have been screened using traditional methods. Traditional methods offer limited insight into the attraction mechanisms in insects, which has limited the development of effectively applied PVOCs and attractants in the field. In addition, most screened PVOCs have been sourced from host plants, leaving the effects of known attractants from non-host plants (i.e., floral volatiles) unexplored. Because unique combinations of non-host and background PVOCs exist in field systems, the potential confounding effects of background odorants on the efficacy of odorant insect trapping remain unknown. Future work must include the analysis of the olfactory molecular mechanism of non-host volatiles in the development of new attractant applications and technologies. With the previous success of PVOC pest management systems and the increased understanding of insect olfactory systems, enabling improved PVOC-screening technologies and leveraging insect olfaction for sustainable pest management will be an essential piece of future sustainable pest management worldwide. 

## Figures and Tables

**Figure 1 plants-13-00185-f001:**
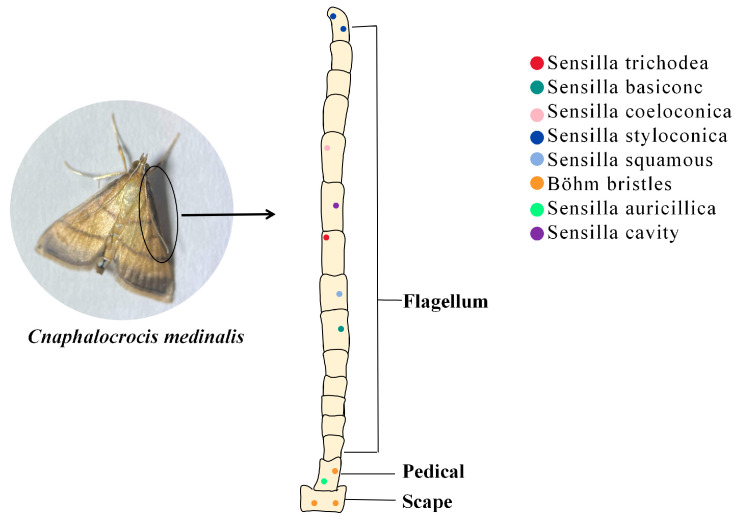
The role of insect antennal olfactory sensors in the recognition of PVOCs. Sensilla trichodea (red dot): mainly distributed in the center of the ventral surface of the antenna; Sensilla basiconc (dark green dot): distributed on the ventral and dorsal surface; Sensilla coeloconica (pink dot): distributed on the ventral surface; Sensilla styloconica (blue dot): distributed on the ventral surface and the end of the antennae; Sensilla squamous (light blue dot): distributed on the dorsal surface; Böhm bristles (orange dot): distributed on the scape and pedicel; Sensilla auricillica (green dot): distributed on the scape; Sensilla cavity (purple dot): distributed on all antenna surfaces.

**Figure 2 plants-13-00185-f002:**
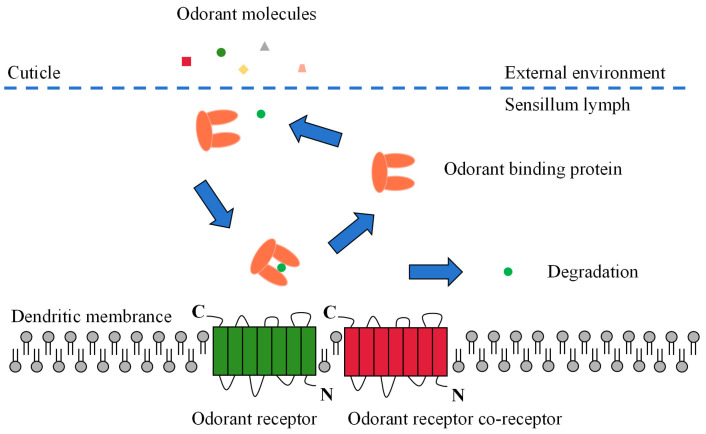
Schematic view of the odorant perception process in insects.

**Figure 3 plants-13-00185-f003:**
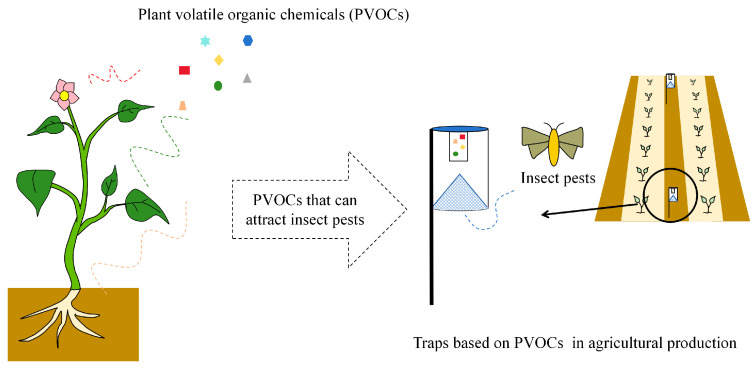
Utilization of pest attractants based on PVOCs.

**Table 1 plants-13-00185-t001:** Plant volatile organic chemicals [12,13].

Plant Organs	Types of Volatiles
Green leaf	C6/C9 aldehydes, alcohols, and esters
Flower	Terpenoids, phenylpropanoids/benzenoids, and fatty acid derivatives
Root	Alcohols, ketones, aldehydes, esters, terpenes, furans, organic acids, aromatic compounds, and sulfur compounds

**Table 2 plants-13-00185-t002:** Olfactory-related gene information in antennae of *Cnaphalocrocis medinalis* and functions of olfactory-related proteins.

Gene Name	GenBank ^a^	Functions of Olfactory-Related Proteins
*CmedPBP4*	KC507185.1	(*Z*)-13-octadecenyl acetate, (*Z*)-11-hexadecenal, (*Z*)-13-octadecenol, cyclohexanol, nerolidol, cedrol, dodecanal, ionone, (−)-α-cedrene, (*Z*)-farnesene, β-myrcene, *R*-(+)-limonene, (−)-limonene, (+)-3-carene [69]
*CmedOBP14*	KP975125.1	cedrol, β-ionone, nerolidol, 3-carene, 1-octen-3-ol, tetradecane, *D*-limonene, 3-pentanol, *L*-limonene, *P*-cymene, (*E*)-2-hexen-1-ol, 2-heptanone, (Z)-hex-3-en-1-ol, hexadecane, cyclohexanol, linalool, 2-tridecanone, henicosane, heptan-1-ol, α-cedrene, octadecane, sabinene, dodecanal, eicosane, nonadecane, *trans*-2-hexenal [70]
*CmedCSP1*	KC507178.1	(+/−)-linalool, nerolidol, tetradecane, nerolidol, cedrol, *cis*-β-farnesene, terpinene-4-ol, α-terpineol, α-terpinene, β-myrcene, sabinene, *P*-cymene, methyl benzoate, γ-terpinene, heptano, *R*-(+)-limonene, 2-heptanone, (1*R*)-(+)-α-pinene [72]
*CmedCSP2*	KC507180.1	cyclohexanol, heptadecane, octadecane, nonadecane, eicosane, (−)-α-cedrene, nerolidol, cedrol, *cis*-β-farnesene, α-terpineol, α-terpinene, β-myrcene, sabinene, *P*-cymene, methyl benzoate, (−)-(*E*)-caryophyllene, γ-terpinene, octane, ionone, 2-heptanone [72]
*CmedCSP3*	KC507182.1	octadecane, (−)-α-cedrene, *cis*-β-farnesene, (−)-(*E*)-caryophyllene, (*Z*)-11-hexadecenyl acetate, (*Z*)-11-hexadecenal [72]
*CmedCSP33*	KP975096.1	2-heptanone, β-ionone, *R*-(+)-limonene, cyclohexanol, (*E*)-2-hexen-1-ol, 3-pentanol, nerolidol [71]

Note: ^a^ Data from National Center for Biotechnology Information website (https://www.ncbi.nlm.nih.gov/, accessed on 1 October 2023).

## Data Availability

No new data were created or analyzed in this study. Data sharing is not applicable to this article.

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
