# Peer review of "The Plant Volatile-Sensing Mechanism of Insects and Its Utilization"

_plants, 2024, doi:10.3390/plants13020185_

Round 1

Reviewer 1 Report

Comments and Suggestions for Authors

The paper has an interesting topic and it is well-presented.

Author Response

The paper has an interesting topic and it is well-presented.

Response: Thank you for your endorsement.

Reviewer 2 Report

Comments and Suggestions for Authors

The authors have developed and interesting and updated review, about the sensing mechanism of insects to plant volatile organic chemicals (PVOCs). They develop the increasing applications of the understanding of this sensing mechanism, about the pest management in agriculture.

The interest of the review for the readers of the Plants journal, it would be increased with the incorporation of additional colour Figures, which provide a better understanding of the concepts of the review, and also it could report an impact and informative picture of the topics of the review. That could improve the interest for readers and the citations of the review.

Alternatively, the colour Figures can be available for readers as electronic supplementary material, from the website of the journal.

I recommend the elaboration of Figures in the following Sections of the review, that could act as a graphical abstract of each Section:

2. Plant volatile organic chemicals. (Figure 1).

3.1. The role of insect antennal olfactory sensors in the recognition of PVOCs. (Figure 2).

3.2. The role of insect antennal olfaction-related proteins in the recognition of PVOCs. (Figure 3, current and only Figure 1).

4. The attraction mechanism of plant volatiles to insects: a case study of C. medinalis. (Figure 4).

5. Development and utilization of pest attractants based on PVOCs. (Figure 5).

If the authors require assistance for the elaboration of the new Figures, I recommend the use of https://www.biorender.com/, or equivalent graphic tools of scientific topics.

Best regards.

Author Response

The authors have developed and interesting and updated review, about the sensing mechanism of insects to plant volatile organic chemicals (PVOCs). They develop the increasing applications of the understanding of this sensing mechanism, about the pest management in agriculture.

Response: Thank you for your endorsement.

The interest of the review for the readers of the Plants journal, it would be increased with the incorporation of additional colour Figures, which provide a better understanding of the concepts of the review, and also it could report an impact and informative picture of the topics of the review. That could improve the interest for readers and the citations of the review.

Response: Thank you very much for your comments, which are helpful in the improvement of our manuscript. We have corrected these issues as recommended.

Alternatively, the colour Figures can be available for readers as electronic supplementary material, from the website of the journal.

I recommend the elaboration of Figures in the following Sections of the review, that could act as a graphical abstract of each Section:

  1. Plant volatile organic chemicals.(Figure 1).

3.1. The role of insect antennal olfactory sensors in the recognition of PVOCs. (Figure 2).

3.2. The role of insect antennal olfaction-related proteins in the recognition of PVOCs. (Figure 3, current and only Figure 1).

  1. The attraction mechanism of plant volatiles to insects: a case study of C. medinalis. (Figure 4).
  2. Development and utilization of pest attractants according to PVOCs.(Figure 5).

If the authors require assistance for the elaboration of the new Figures, I recommend the use of https://www.biorender.com/, or equivalent graphic tools of scientific topics.

Best regards.

Response: Thank you for your valuable comments, which have assisted us in improving our manuscript. We have included corresponding figures in the revised version and added two tables inPlant volatiles section and “The attraction mechanism of plant volatiles to insects: a case study of C. medinalis” section.

Reviewer 3 Report

Comments and Suggestions for Authors

plants-2666562-peer-review-v1 -Review

The Sensing Mechanism of Insects to Plant Volatiles and Its Utilization

1-      In this review, the authors summarize past and recent advances of how insects recognize and utilize plant volatiles.

A time range, and the databases that were used, could be mentioned following the aforementioned paragraph.

2-      A paragraph commenting on the devices by which these compounds are used should be included in this paragraph(Lines 309-319, page 7.

2.1- As they are included in the baits, are they solutions in a certain solvent?, in some polymer, in another device?

2.2- What is the estimated time of effectiveness in days or weeks?.

2.3- Were the compounds used in laboratory or field tests?

3-      Conclusion

Lines 354-355, page 8

The mechanisms of interaction between insect odorant-binding proteins and PVOCs remain to be further explored and verified.

Are there some studies or others through molecular modeling, docking or others on the possible interaction mechanism?

It should be mentioned here, in a brief  lines

4-      Acknowledgments: We are very grateful to all the reviewers for their valuable advice and suggestions to improve the manuscript.

This thank you paragraph is not usual in MDPI Journals, and it is not necessary. Please include your acknowledgments according to the suggestions in the Journal and to Authors instructions.

After the suggestions made, the paper should be considered for publication

Comments on the Quality of English Language

Minor editing of English language required

Author Response

plants-2666562-peer-review-v1 -Review

The Sensing Mechanism of Insects to Plant Volatiles and Its Utilization

1-      In this review, the authors summarize past and recent advances of how insects recognize and utilize plant volatiles.

A time range, and the databases that were used, could be mentioned following the aforementioned paragraph.

Response: This sentence (Lines 55-56) has been modified as follows: “In this review, we employ NCBI PubMed to summarize the progress in comprehending how insects recognize and utilize plant volatiles over the past decade.

2-      A paragraph commenting on the devices by which these compounds are used should be included in this paragraph(Lines 309-319, page 7).

Response: These sentences (Lines 321-327) have been modified as follows: “Using funnel traps and floral volatiles, including methyl salicylate, phenylacetaldehyde, and eugenol diluted in n-hexane as attractants and replaced every two weeks can trap Cydalima perspectalis adults [78]. Meagher [79] used different floral volatile treatments, including phenylacetaldehyde, benzyl acetate combined with phenylacetaldehyde, benzyl acetate, and benzaldehyde, as bait with standard Universal Moth Traps, ‘Unitraps,’ with insecticide strips to kill moths captured in peanut fields.”

2.1- As they are included in the baits, are they solutions in a certain solvent?, in some polymer, in another device?

Response: This sentence (Lines 321-323) has been revised as follows: “Using funnel traps and floral volatiles, including methyl salicylate, phenylacetaldehyde, and eugenol diluted in n-hexane as attractants and replaced every two weeks can trap Cydalima perspectalis adults [78].”

2.2- What is the estimated time of effectiveness in days or weeks?.

Response: This sentence (Lines 321-323) has been revised as follows: “Using funnel traps and floral volatiles, including methyl salicylate, phenylacetaldehyde, and eugenol diluted in n-hexane as attractants and replaced every two weeks can trap Cydalima perspectalis adults [78].”

2.3- Were the compounds used in laboratory or field tests?

Response: This sentence (Lines 321-323) has been revised as follows: “Using funnel traps and floral volatiles, including methyl salicylate, phenylacetaldehyde, and eugenol diluted in n-hexane as attractants and replaced every two weeks can trap Cydalima perspectalis adults [78].”

3-      Conclusion

Lines 354-355, page 8

The mechanisms of interaction between insect odorant-binding proteins and PVOCs remain to be further explored and verified.

Are there some studies or others through molecular modeling, docking or others on the possible interaction mechanism? It should be mentioned here, in a brief lines

Response: Thank you for this suggestion. The sentence (Lines 367-369) has been modified as follows: “While homology modeling and molecular docking can be utilized to predict the interaction between insect odorant-binding proteins and PVOCs, further exploration and verification through experiments are still required.”

4-      Acknowledgments: We are very grateful to all the reviewers for their valuable advice and suggestions to improve the manuscript.

This thank you paragraph is not usual in MDPI Journals, and it is not necessary. Please include your acknowledgments according to the suggestions in the Journal and to Authors instructions.

Response: Thank you for identifying. We have removed this sentence in the revised manuscript.

After the suggestions made, the paper should be considered for publication.

Response: Thank you for this feedback. Thank you very much for your comments, which have helped us to improve our manuscript. We have corrected these issues as recommended.